# Heat Treatment Optimization of 2219 Aluminum Alloy Fabricated by Wire-Arc Additive Manufacturing

**Jiannan Yang** [1,2], **Yunqiang Ni** [3], **Hui Li** [2], **Xuewei Fang** [1,2,*] and **Bingheng Lu** [1,2]

1   The State Key Laboratory for Manufacturing Systems Engineering, Xi'an Jiaotong University, Xi'an 710054, China
2   National Innovation Institute of Additive Manufacturing, Xi'an 710065, China
3   Weichai Holding Group Co., Ltd., Wei Fang 261061, China
*   Correspondence: xueweifang@xjtu.edu.cn; Tel.: +86-151-0298-5026 or +86-29-8339-5028

**Abstract:** Wire-arc additive manufacturing has generated significant interest in the aerospace industry for the fabrication of large aluminum alloy components such as alloy 2219 (Al-6.3Cu). However, its application is limited by the low strength of the deposited parts. In this study, the effect of heat-treatment parameters, including solution temperature, solution time, aging temperature, and aging time, on the mechanical properties was optimized by using the Taguchi method. The results show that the solution temperature is the most influential factor on ultimate tensile strength and yielding strength, while the aging time had the most significant effect on elongation. Thereafter, the best control factor for the maximum response variable was obtained. Microhardness and strength properties were greatly improved after optimized T6 heat treatment. The strengthening mechanism of this additively fabricated Al-6.3Cu alloy was investigated by microstructural analysis.

**Keywords:** 2219 aluminum alloy; wire-arc additive manufacturing; heat treatment; microstructure; mechanical property





## 1. Introduction

Additive manufacturing (AM) is a novel and promising technology that is used to fabricate a near-net-shape part layer by layer, resulting in mechanical and tribological properties that differ from those of conventionally manufactured parts [1,2]. As a powerful tool, AM is expected to assist Industry 4.0 by providing freedom in geometry, material design, quality, and logistics [3,4]. Depending on the type of the feedstock unit, metal additive manufacturing techniques can be classified as powder feeding and wire feeding [5–8]. Among the metal additive manufacturing methods, wire-arc additive manufacturing (WAAM) technology has now expanded the metal AM market due to its high deposition rates and low construction and investment costs for manufacturing custom and large metal parts, ranging from a few kilograms to several tons [9].

The WAAM process uses metal wire as a raw material and melts it to deposit material layer by layer through an electric arc, using such methods as gas metal arc welding (GMAW), gas tungsten arc welding (GTAW), and plasma arc welding (PAW) [10–12]. In order to improve arc stability and reduce the splash of the GMAW process, Fronius invented an improved welding process called cold metal transfer (CMT) [13,14]. The CMT process is a relatively new process, but it is fit for aluminum alloys because of its advantages of low heat input, small distortion, and a high deposition rate [15–18].

High-strength aluminum alloys are becoming increasingly important in the aerospace and automotive industries, due to their light weight. Al-Cu alloy, especially 2219 alloy, is a promising material for aerospace requirements, such as the requirements for airframes or spacecraft enclosures. However, the ultimate tensile strength (about 283 MPa) of WAAM 2219 alloy is lower than required [18]. For production applications, the strength of the WAAM

aluminum alloy needs to be increased to a level close to its forging counterpart through WAAM process optimization or through auxiliary methods such as interlayer hammering [19].

Since 2219 aluminum alloy is a kind of precipitation-hardening alloy, dispersion hardening through heat treatment is a common choice [20]. Solution treatment and aging treatment (T6 tempering) are usually performed to improve the strength. Raza et al. [21] worked on the effect of aging temperature and aging time on the mechanical properties and microstructure of 2219 aluminum alloy. Tiryakioglu et al. [22] studied the quenching susceptibility of 2219-T87 aluminum alloy. However, there are no relevant reports on the effects of solid solution temperature and solution time on the microstructure and mechanical properties of 2219 aluminum alloy. Therefore, this paper aims to systematically optimize the post-heat treatment process and investigate the strengthening mechanism for WAAM 2219 aluminum alloy. Regarding the mechanical properties of WAAM 2219 aluminum alloy, orthogonal experiments were conducted to optimize the major process parameters, including solution temperature, solution time, aging temperature, and aging time. This work will further the understanding of the heat-treatment strengthening of WAAM aluminum alloys and facilitate their production applications in the future.

## 2. Materials and Methods

### 2.1. Preparation of Deposits

The chemical compositions of the ER2319 wire and 2219-T87 plate are detailed in Table 1. A Fronius CMT Advanced 4000 R was used as the heat source in this study. As described in our previous study [19], the use of pulse-advanced cold metal transfer (CMT-PADV) allowed the production of a near-porosity-free deposition of WAAM 2219 alloy and refined equiaxed grains. Therefore, in this study, the CMT-PADV mode was applied to construct wall-shaped parts. Table 2 shows the primary deposition parameters: wire feed speed of 7 m/min and travel speed of 0.5 m/min. Pure argon (99.999%) was set at a flow rate of 25 L/min to protect the molten pool and the deposited part. The distance between the contact head and the workpiece was kept at 15 mm.

**Table 1.** Chemical composition of ER2319 wire and 2219-T87 plate.

| Alloys | Chemical Composition (wt.%) | | | | | | | | |
|---|---|---|---|---|---|---|---|---|---|
| | Cu | Mn | Mg | Ti | Zr | V | Zn | Si | Fe |
| ER2319 | 5.8–6.8 | 0.2–0.4 | $\leqq$0.02 | 0.1–0.2 | 0.1–0.25 | 0.05–0.15 | $\leqq$0.1 | $\leqq$0.2 | $\leqq$0.3 |
| 2219-T87 | 5.8–6.8 | 0.2–0.4 | $\leqq$0.02 | 0.02–0.1 | 0.1–0.25 | 0.05–0.15 | $\leqq$0.1 | $\leqq$0.2 | $\leqq$0.3 |

**Table 2.** Primary deposition parameters for wall-shaped structure of 2219 aluminum alloy.

| WAAM Parameters | |
|---|---|
| Arc mode | CMT-PADV mode |
| Wire diameter | 1.2 mm |
| Wire feed speed | 7 m/min |
| Travel speed | 0.5 m/min |
| Shielding gas flow rate | 25 L/min (99.999%) |

### 2.2. Orthogonal Experiment Design

Solid solution treatment plus artificial aging (T6) were conducted to improve the performance of the deposition. Taguchi's method was used to design experiments to optimize the T6 heat treatment process parameters, including solution temperature, solution time, aging temperature, and aging time. Ultimate tensile strength (UTS), yield strength (YS), and elongation were chosen as indicators. Samples for tensile testing were cut, as shown in Figure 1. The tensile tests were performed following the ISO 6892-1-2009 standard and carried out at ambient temperature with a constant strain rate of 0.005 min$^{-1}$. Analysis

of variance (ANOVA) was applied to the statistical treatment of the experimental results to predict the contribution of each control variable to a given level, using Minitab 17 software.

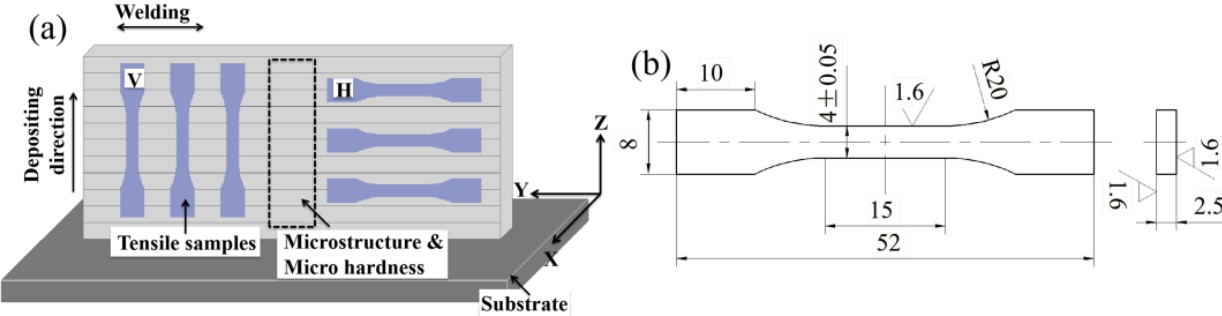

**Figure 1.** Schematics of (**a**) the sampling positions and (**b**) the dimensions of tensile samples.

*2.3. Microstructure Observation*

Microstructural investigations were conducted on the cross sections of the wall-shaped part after T6 heat treatment. Optical microscopy (OM) (DM2700M, LEICA, Wetzlar, German), scanning electron microscopy (SEM) (SU3500, HITACHI, Tokyo, Japan), electron backscattered diffraction (EBSD) (FE-SEM JSM-7900F, JEOL, Tokyo, Japan) equipped with Oxford EBSD system (Oxford Instruments, Oxford, UK), and transmission electron microscopy (TEM) (JEM-2100F, JEOL, Tokyo, Japan) were utilized to obtain the microstructure at different scales. Due to the requirements in OM and SEM testing, Kroll's reagent was used to etch the specimens. TEM foil specimens with a diameter of 3 mm were mechanically polished to approximately 200 µm and double-jet electropolished with a 30% nitric acid and 70% methanol solution at −30 °C and 15 V. XRD phase analysis was conducted through an X-ray diffractometer (D8 Advanced, BRUKER, Massachusetts, USA). The scanning angle ranged from 20° to 90°. Vickers microhardness test was performed at a load of 200 g for 15 s.

**3. Results**

*3.1. Orthogonal Experiment Analysis*

3.1.1. ANOVA

Each experiment was repeated three times. Table 3 summarizes the average experimental results for UTS, YS, and elongation as response variables. ANOVA analysis was then employed to determine how different factors affected the response variables. The results of ANOVA for the UTS, YS, and elongation are shown in Table 4a–c. $\alpha = 0.05$ (confidence level of 95%) was taken as the significance level. Table 4a–c shows the significance levels achieved for UTS, YS, and elongation associated with the F-test for each source of variation, respectively. A larger F-value of a variable indicates a greater effect on the performance characteristics due to changes in that process variable. When the *p*-value was lower than 0.05, the effect of the source on the response was thought to be statistically significant at the 95% confidence level.

In Table 4a,b, the results show that the F-values of solution temperature are greater than those of the other three factors, i.e., the largest contribution to UTS and YS is caused by solution temperature. The effects of all variables except aging temperature were statistically significant (*p*-value < 0.05). Solution temperature, solution time and aging time contributed 74.67%, 10.27%, and 10.64%, respectively. As shown in Table 4b, the solution time shows a statistical insignificance on YS (*p*-value > 0.05). Solution temperature, aging temperature, and aging time contributed 39.54%, 17.44%, and 39.08%, respectively. In Table 4c, the ANOVA results show that only the effect of aging time on elongation is statistically significant. The contribution of aging time to the elongation is the largest. Solution temperature, solution time, aging temperature and aging time contributed 1.91%, 5.21%, 30.79%, and 56.76%, respectively.

**Table 3.** L16 orthogonal array experiment design and responses.

| NO. | Control Variables | | | | Average of Responses | | |
|---|---|---|---|---|---|---|---|
| | Solution Temperature (°C) | Solution Time (min) | Aging Temperature (°C) | Aging Time (h) | UTS (MPa) | YS (MPa) | Elongation (%) |
| 1 | 515 | 30 | 155 | 1 | 328.11 (5.92) | 177.07 (3.88) | 16.93 (0.28) |
| 2 | 515 | 60 | 165 | 6 | 382.65 (4.05) | 221.74 (2.62) | 12.90 (0.04) |
| 3 | 515 | 90 | 175 | 11 | 389.03 (12.43) | 272.40 (0.02) | 8.83 (0.71) |
| 4 | 515 | 120 | 185 | 16 | 407.88 (11.61) | 299.66 (5.37) | 7.76 (1.50) |
| 5 | 525 | 30 | 165 | 11 | 408.01 (11.20) | 262.79 (4.26) | 10.31 (0.84) |
| 6 | 525 | 60 | 155 | 16 | 417.03 (6.03) | 259.15 (3.76) | 13.07 (1.41) |
| 7 | 525 | 90 | 185 | 1 | 405.41 (6.06) | 242.48 (5.17) | 16.80 (1.51) |
| 8 | 525 | 120 | 175 | 6 | 431.60 (7.46) | 273.47 (14.23) | 8.78 (2.07) |
| 9 | 535 | 30 | 175 | 16 | 430.73 (5.83) | 302.16 (2.33) | 9.05 (0.65) |
| 10 | 535 | 60 | 185 | 11 | 435.07 (8.72) | 309.85 (1.85) | 9.04 (1.31) |
| 11 | 535 | 90 | 155 | 6 | 425.23 (9.70) | 255.08 (5.72) | 15.31 (1.07) |
| 12 | 535 | 120 | 165 | 1 | 430.79 (8.36) | 246.45 (7.30) | 17.89 (1.00) |
| 13 | 545 | 30 | 185 | 6 | 450.44 (8.55) | 329.71 (10.37) | 6.44 (1.25) |
| 14 | 545 | 60 | 175 | 1 | 441.64 (2.81) | 267.64 (3.76) | 15.33 (1.10) |
| 15 | 545 | 90 | 165 | 16 | 462.27 (5.01) | 322.95 (6.36) | 10.04 (0.80) |
| 16 | 545 | 120 | 155 | 11 | 465.74 (10.72) | 316.23 (14.27) | 14.78 (1.37) |

**Table 4.** Variance analysis of means for (a) UTS, (b) YS, and (c) elongation.

| Source | DF | Adj SS | Adj MS | F | P | Contribution (%) |
|---|---|---|---|---|---|---|
| **(a)** | | | | | | |
| Solution temperature | 3 | 12,844.1 | 4281.37 | 87.01 | 0.002 | 74.67 |
| Solution time | 3 | 1767.2 | 589.08 | 11.97 | 0.035 | 10.27 |
| Aging temperature | 3 | 611.3 | 203.76 | 4.14 | 0.137 | 3.55 |
| Aging time | 3 | 1830.0 | 610.01 | 12.40 | 0.034 | 10.64 |
| Error | 3 | 147.6 | 49.21 | - | - | 0.86 |
| Total | 15 | 17,200.3 | - | - | - | - |
| **(b)** | | | | | | |
| Solution temperature | 3 | 9730.1 | 3243.37 | 89.27 | 0.002 | 39.54 |
| Solution time | 3 | 860.7 | 286.90 | 7.90 | 0.062 | 3.50 |
| Aging temperature | 3 | 4291.9 | 1430.65 | 39.38 | 0.007 | 17.44 |
| Aging time | 3 | 9615.2 | 3205.08 | 88.22 | 0.002 | 39.08 |
| Error | 3 | 109.0 | 36.33 | - | - | 0.44 |
| Total | 15 | 24,607.0 | - | - | - | - |
| **(c)** | | | | | | |
| Solution temperature | 3 | 3.958 | 1.319 | 0.36 | 0.789 | 1.91 |
| Solution time | 3 | 10.799 | 3.600 | 0.98 | 0.507 | 5.21 |
| Aging temperature | 3 | 63.778 | 21.259 | 5.78 | 0.092 | 30.79 |
| Aging time | 3 | 117.570 | 39.190 | 10.66 | 0.042 | 56.76 |
| Error | 3 | 11.031 | 3.677 | - | - | 5.33 |
| Total | 15 | 207.136 | - | - | - | - |

### 3.1.2. Evaluation of Means and *S/N* Ratios for Optimization Design

Signal-to-noise ratio (*S/N*) and average response data are Taguchi's highly recommended methods for completion analysis of multiple runs. Depending on the needs of practical industrial production, the *S/N* ratio is divided into smaller-better type features, larger-better type features, and nominal-better type features. Since maximization is the object of this work, the *S/N* ratio is defined in accordance with the Taguchi method:

$$S/N = -10 \log_{10}[1/n \sum_{1}^{n} 1/Y_i^2] \tag{1}$$

where $Y_i$ is the eigenvalue, *i* is the number of observations, and *n* is the number of replicates.

The average values of mean and *S/N* for UTS, YS, and elongation are provided in Tables 5–7 and plotted in Figures 2–4. A higher level of *S/N* ratio indicates a better overall performance. Based on the *S/N* ratio and ANOVA, the optimized control variables for UTS and YS were at level 4 for all variables (Tables 5 and 6). The optimized control variables for elongation were the solution temperature at level 3, the solution time at level 2, the aging temperature at level 1, and the aging time at level 1 (Table 7). The rankings based on the Delta statistic in Tables 4–6 compared the relative magnitudes of impacts. The rankings were assigned based on the Delta values. The descending order of the UTS ranking was solution temperature > solution time > aging time > aging temperature. The descending order of the YS ranking was solution temperature > aging time > aging temperature > solution time. The descending order of elongation ranking was aging time > aging temperature > solution time > solution temperature. As shown in Figures 2 and 3, it is clear that the solution temperature had the greatest effect, contributing 74.67% and 39.54% to UTS and YS, respectively. There was a significant increase in both UTS and YS as the solution temperature increased. Table 8 summarizes the values of the corresponding control variables to obtain the optimized UTS, YS, and elongation.

**Table 5.** Response table for signal-to-noise ratios (larger is better) and means for UTS.

| Level | *S/N* Data | | | | Mean Data | | | |
|---|---|---|---|---|---|---|---|---|
| | Solution Temperature | Solution Time | Aging Temperature | Aging Time | Solution Temperature | Solution Time | Aging Temperature | Aging Time |
| 1 | 51.50 | 52.07 | 52.16 | 52.02 | 376.9 | 404.3 | 409.0 | 401.5 |
| 2 | 52.37 | 52.43 | 52.46 | 52.50 | 415.5 | 419.1 | 420.9 | 422.5 |
| 3 | 52.68 | 52.46 | 52.52 | 52.54 | 430.5 | 420.5 | 423.3 | 424.5 |
| 4 | 53.16 | 52.74 | 52.55 | 52.65 | 455.0 | 434.0 | 424.7 | 429.5 |
| Delta | 1.66 | 0.67 | 0.39 | 0.63 | 78.1 | 29.7 | 15.7 | 28.0 |
| Rank | 1 | 2 | 4 | 3 | 1 | 2 | 4 | 3 |

**Table 6.** Response table for signal-to-noise ratios (larger is better) and means for YS.

| Level | *S/N* Data | | | | Mean Data | | | |
|---|---|---|---|---|---|---|---|---|
| | Solution Temperature | Solution Time | Aging Temperature | Aging Time | Solution Temperature | Solution Time | Aging Temperature | Aging Time |
| 1 | 47.53 | 48.33 | 47.84 | 47.26 | 242.7 | 267.9 | 251.9 | 233.4 |
| 2 | 48.27 | 48.39 | 48.33 | 48.54 | 259.5 | 264.6 | 263.5 | 270.0 |
| 3 | 48.85 | 48.68 | 48.90 | 49.23 | 278.4 | 273.2 | 278.9 | 290.3 |
| 4 | 49.77 | 49.03 | 49.35 | 49.40 | 309.1 | 284.0 | 295.4 | 296.0 |
| Delta | 2.24 | 0.70 | 1.51 | 2.14 | 66.4 | 19.4 | 43.5 | 62.6 |
| Rank | 1 | 4 | 3 | 2 | 1 | 4 | 3 | 2 |

**Table 7.** Response table for signal-to-noise ratios (larger is better) and means for elongation.

| Level | S/N Data | | | | Mean Data | | | |
|---|---|---|---|---|---|---|---|---|
| | Solution Temperature | Solution Time | Aging Temperature | Aging Time | Solution Temperature | Solution Time | Aging Temperature | Aging Time |
| 1 | 20.88 | 20.04 | 23.50 | 24.46 | 11.605 | 10.683 | 15.023 | 16.728 |
| 2 | 21.49 | 21.84 | 21.89 | 20.24 | 12.240 | 12.585 | 12.785 | 10.858 |
| 3 | 21.75 | 21.79 | 20.16 | 20.43 | 12.823 | 12.745 | 10.498 | 10.740 |
| 4 | 20.83 | 21.28 | 19.40 | 19.82 | 11.647 | 12.303 | 10.010 | 9.980 |
| Delta | 0.92 | 1.81 | 4.10 | 4.64 | 1.218 | 2.063 | 5.013 | 6.758 |
| Rank | 4 | 3 | 2 | 1 | 4 | 3 | 2 | 1 |

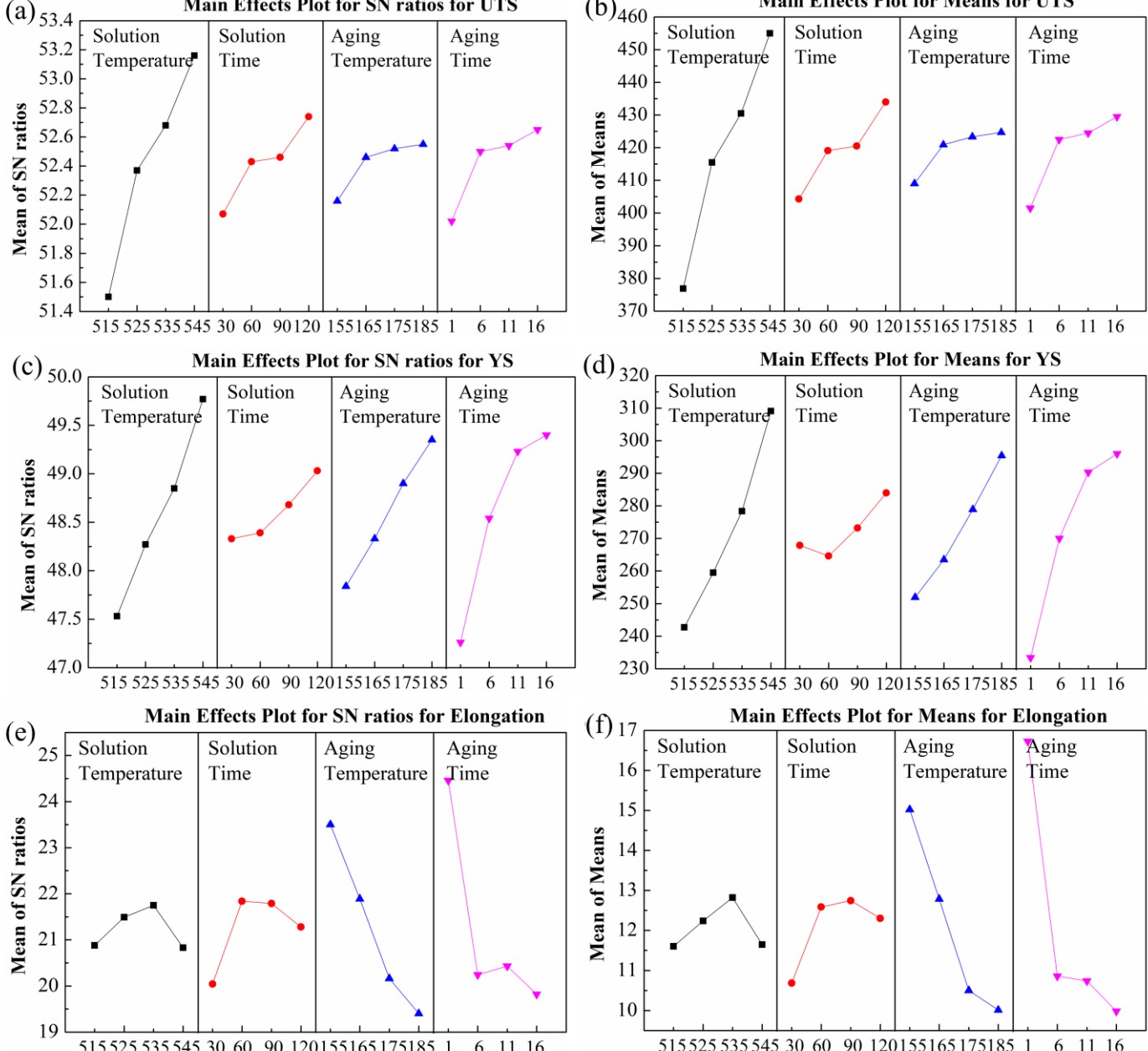

**Figure 2.** Main effect plots for *S/N* ratio and means for (**a**,**b**) UTS, (**c**,**d**) YS, and (**e**,**f**) elongation.

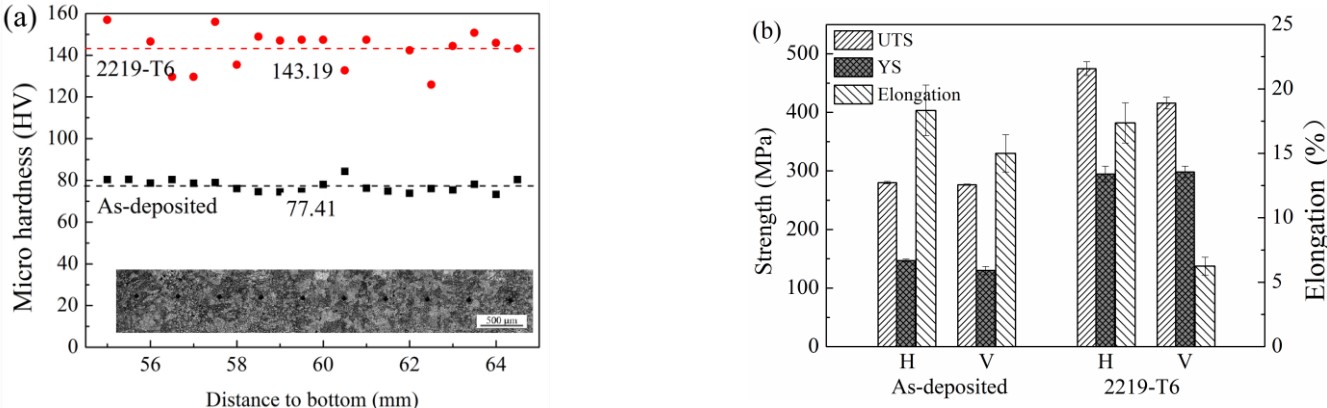

**Figure 3.** (**a**) The micro hardness distribution and (**b**) tensile properties for the as-deposited and heat-treated WAAM 2219 alloy.

**Figure 4.** (**a**) The sampling positions of the wall; OM images of WAAM 2219 alloys (**b**) as-deposited and (**c**) T6 heat-treated.

**Table 8.** Optimized values of control variables for different response variables.

| Control Variables | Optimized Values for Response Variables | | |
| --- | --- | --- | --- |
| | UTS | YS | Elongation |
| Solution temperature (°C) | 545 | 545 | 535 |
| Solution time (min) | 120 | 120 | 60 |
| Aging temperature (°C) | 185 | 185 | 155 |
| Aging time (h) | 16 | 16 | 1 |

*3.2. Properties of Wire and Arc Additively Manufactured 2219 Aluminum Alloy with T6 Heat Treatment*

3.2.1. Mechanical Properties

According to the results of the orthogonal experiments, in order to balance the three different performance indices (UTS, YS, and elongation), the solid solution treatment temperature of WAAM 2219 aluminum alloy was set to be held at 535 °C for 60 min, followed by water quenching and, then, artificial aging treatment held at 175 °C for 6 h. Figure 3 shows the microhardness distribution and tensile properties of the WAAM 2219 alloy after deposition and heat treatment. As can be seen in Figure 3a, the average value for alloy 2219-T6 was 143.19 HV, which represented an 85% increase in hardness compared to the hardness of as-deposited sample. Although the microhardness of 2219-T6 alloy fluctuated among different testing points, each test point greatly exceeded the hardness value of the as-deposited alloy. Figure 3b shows the UTS, YS, and elongation test results for the as-deposited and heat-treated samples. The UTS and YS of as-deposited 2219 aluminum alloy were 261.3 MPa and 118.6 MPa, which were significantly lower than the strength of the 2219-T6 alloy (UTS: 441.85 MPa; YS: 293.9 MPa). However, the elongation result of T6 heat-treated WAAM 2219 alloy was slightly lower than that of the as-deposited alloy. It can be concluded that T6 heat treatment is a desirable process to significantly improve the mechanical properties of as-deposited 2219 aluminum alloy.

3.2.2. Microstructure

The microstructure of WAAM 2219 aluminum alloy is complex. The morphology of the solidification structure is determined by the temperature gradient and the cooling rate of the molten pool. According to the traditional welding metallurgy theory [23,24], the microstructure of WAAM 2219 aluminum alloy prepared on the basis of the welding process can be divided into an accumulation zone, a fusion zone, and a heat-affected zone. The zones are divided as shown in the model in Figure 4a. The accumulation layer of each layer is a complete melting zone, and the previous accumulation layer will become the heat-affected zone of the next accumulation; there will be a narrow partial melting zone between every two layers. Figure 4b shows the OM image of the as-deposited WAAM 2219 alloy (location of the red rectangle in Figure 4a). The relatively low heat input of the CMT-PA process resulted in a relatively homogeneous microstructure, including fine dendrites, fine equiaxed crystals, and a small number of columnar dendrites [19]. After the T6 heat treatment, the microstructure of the as-deposited sample tended to be more homogeneous (Figure 4c). Meanwhile, the number of dendrites contained in the grains was reduced, while the grain size apparently grew.

Figure 5 shows SEM images of the accumulation zone in as-deposited alloy and T6 heat-treated alloy. As shown in Figure 5a, some of the white second-phase particles dispersed along grain boundaries, while some distributed within the grains in the as-deposited alloy. The XRD (Figure 6a) and EDS (Figure 6b) results demonstrated that these white particles were Al-Cu eutectics ($\alpha$-Al and θ phases). Figure 5b shows that the eutectics were evenly distributed in the base matrix of the heat-treated samples. It is noted that when the temperature approached the eutectic temperature (543 °C), most of the eutectics dissolved, leading to the reduced size of second-phase particles after T6 heat treatment.

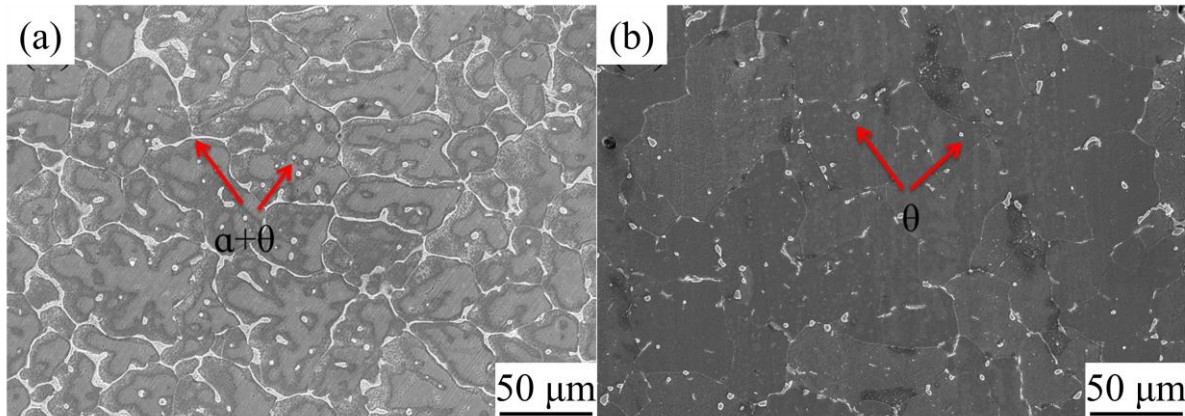

**Figure 5.** SEM images of (**a**) as-deposited and (**b**) T6 heat-treated.

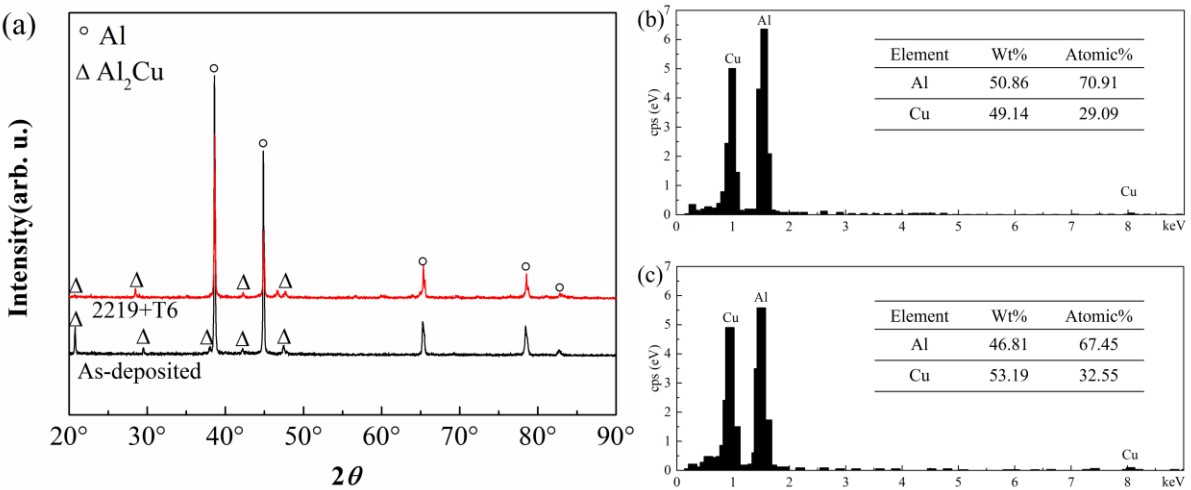

**Figure 6.** (**a**) XRD pattern of WAAM 2219 alloys; EDS analysis for the second phase in the WAAM 2219 alloys (**b**) as-deposited and (**c**) T6 heat-treated.

Figure 7a,b depicts the inverse-polar-map (IPF) EBSD plots of the cross sections (x–z planes) of the as-deposited and T6 heat-treated samples. The microstructure of the as-deposited and T6 heat-treated metal consisted of equiaxed grains. The grain size histograms for different specimens are shown in Figure 7c,d. Given that the resolution limitation of the EBSD scan cannot be neglected, only grains larger than 3 μm were taken into account. It can be seen that the size distribution of the grains was more uniform after heat treatment. Despite the small change in grain size (from 42.1 μm to 40.7 μm), significant improvement in both UTS and YS was observed for the heat-treated 2219 alloy. Most of the stable θ phases dissoluted during the solution process and numerous metastable precipitates formed during the subsequent aging process. Figure 8 shows the TEM images of the T6 heat-treated alloy. The T6 heat treatment samples (quench treatment at 535 °C for 60 min and then age treatment at 175 °C for 6 h) were first cut with 3 mm in diameter and 0.2 mm in thickness, then by mechanical polishing and electro-polishing until the specimens met the TEM observation requirements. As shown in Figure 7a, fine needle-like precipitates were densely and uniformly distributed in the matrix after the T6 treatment. The selected area electron diffraction pattern suggested that these precipitates corresponded to the θ′ phase. For example, the brighter diffraction spot at the center of the red circle corresponded to the (−1 −1 0) plane of the θ′ phase. As pointed out by Huang and Kou, the size and distribution of Al$_2$Cu determine the mechanical properties of 2219 aluminum alloy [25]. These fine precipitates are considered to be the barrier to impede the dislocation movement within the alloy.

Dispersion strengthening is the most important strengthening method in aluminum alloy. The type, size, and shape of the precipitation phase are the main factors determining the strength of aluminum alloy. The precipitation phase can effectively prevent the movement of grain boundaries and dislocations, thereby increasing the strength of the alloy. The aging sequence in Al-Cu alloy is as shown in Equation (2):

$$G.P(I) \rightarrow G.P(II) \rightarrow \theta' \rightarrow \theta \tag{2}$$

In the quenched state, the copper atoms are randomly and chaotically distributed in the matrix. At the beginning of aging, copper atoms accumulate on some crystal planes on the aluminum matrix to form a solute atomic segregation zone, i.e., the G.P(Guinier-Preston) (I) region [26]. Gu et al. [27] found that the average grain size of the 45 kN rolled + T6 treated specimen was almost half that of the T6-treated specimens without inter-layer rolling, but the mechanical properties, such as hardness and strength, were similar for both alloys. It is inferred that grain boundary strengthening is not the main mechanism for the strength increase of the heat-treated 2219 alloy, as the grain sizes of these two types of specimens are similar.

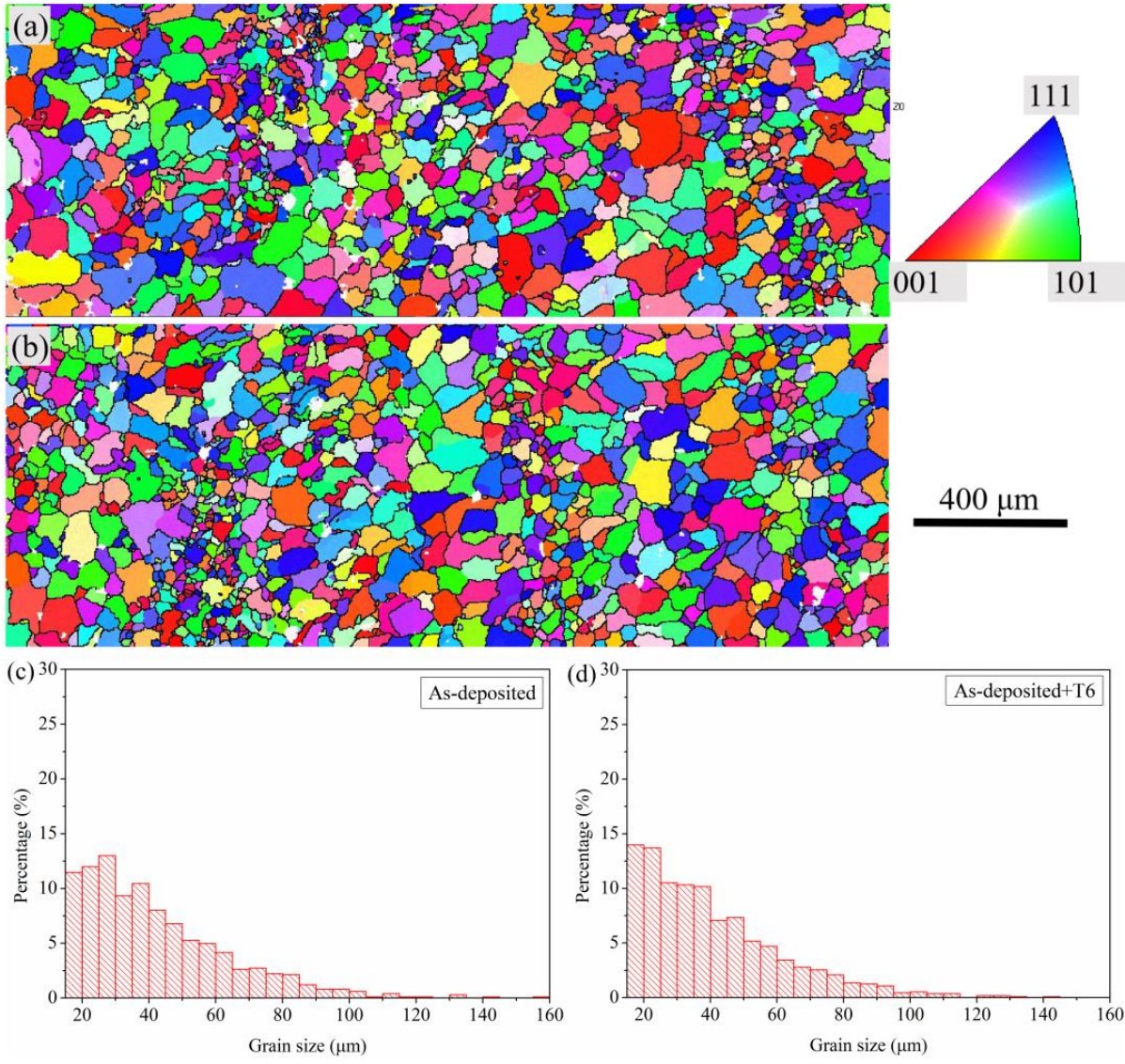

**Figure 7.** IPF orientation maps for (**a**) as-deposited and (**b**) T6 heat-treated alloy; statistics charts of grain size for (**c**) as-deposited and (**d**) T6 heat-treated alloy.

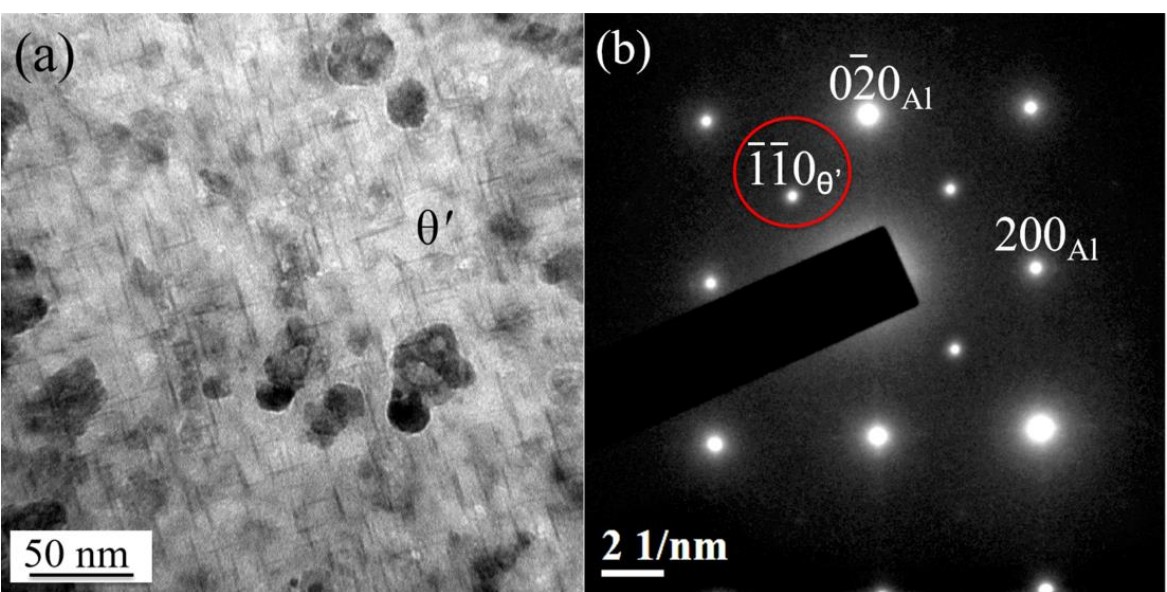

**Figure 8.** TEM images of T6 heat-treated alloy: (**a**) bright field image and (**b**) selected area electron diffraction pattern. The brighter diffraction spot at the center of the red circle corresponded to the (−1 −1 0) plane of the θ′ phase.

## 4. Conclusions

The influences of solution temperature, solution time, aging temperature, and aging time on the tensile performance of WAAM 2219 aluminum alloy were investigated by Taguchi method.

(1) The maximum UTS and YS were obtained at a solution temperature of 545 °C, a solution time of 120 min, an aging temperature of 185 °C, and an aging time of 16 h, and the maximum elongation was obtained at a solution temperature of 535 °C, a solution time of 60 min, an aging temperature of 155 °C, and an aging time of 1 h, where the optimized choice of aging time is very important.

(2) The microhardness and strength properties were greatly improved after the T6 heat treatment. The main strengthening mechanism of this alloy is precipitation strengthening.

(3) The selection of aging time is very important for attaining the maximum elongation. The data from this study contribute to existing databases and will help researchers in the field to build numerical models to obtain the desired mechanical properties.

**Author Contributions:** Conceptualization, J.Y., H.L. and X.F.; methodology, H.L. and X.F.; validation, H.L. and X.F.; formal analysis, H.L.; investigation, H.L.; resources, Y.N. and B.L.; writing—original draft preparation, J.Y. and H.L.; writing—review and editing, J.Y., H.L. and X.F.; supervision, Y.N. and B.L.; project administration, Y.N. and B.L.; funding acquisition, X.F., Y.N. and B.L. All authors have read and agreed to the published version of the manuscript.

**Funding:** The authors would like to acknowledge the financial support from the National Science Foundation of China [Grant No. 52205414]. This work was also supported by the Young Elite Scientists Sponsorship Program by CAST, China [Grant No. 2021QNRC001].

**Institutional Review Board Statement:** Not applicable.

**Informed Consent Statement:** Not applicable.

**Data Availability Statement:** Not applicable.

**Acknowledgments:** Not applicable.

**Conflicts of Interest:** The authors declare no conflict of interest. The funders had no role in the design of the study; in the collection, analyses, or interpretation of data; in the writing of the manuscript; or in the decision to publish the results.

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
