# Peer review of "Heat Treatment Optimization of 2219 Aluminum Alloy Fabricated by Wire-Arc Additive Manufacturing"

_coatings, doi:10.3390/coatings13030610_

Round 1

Reviewer 1 Report

It would be worth considering whether the plate-shaped precipitates Q' in the matrix, whose surfaces are coherent with the matrix and the edges are not (an interfacial boundary is formed - the matrix is the precipitate) have a decisive influence on the nucleation of the Al2Cu phase.

Author Response

Thank you for your positive comments and valuable suggestions to improve the quality of our manuscript entitled “Heat treatment optimization of 2219 aluminum alloy fabricated by wire and arc additive manufacturing” (ID: coatings-2083434) in “Coatings”.

For the comments of our paper, revisions in the Marked-up Revised Manuscript are revised point-by-point, and highlighted in yellow.

Reviewer 2 Report

Title: 

-       The title reads fine and is appealing for a broader research community. 

Abstract: 

-       Please avoid the usage of abbreviations in the abstract since it is a stand-alone part of each article.

-       The following sentence remains unclear “Microhardness and strength properties were greatly 17 prompted after optimal”

-       Please avoid the wording “optimal”. 

Keywords: 

-       The selection of the presented keywords is fine. 

Introduction:

-       Related to the first paragraph, there is a recent review article regarding AM and tribology. Since it belongs to the overall state of the art of the presented article, authors are advised to refer to “Tribological Behavior of Additively Manufactured Metal Components”

-       Please introduce all abbreviations when used for the first time. 

-       The introduction and entire manuscript text should be carefully revised regarding the language used. 

-       The novelty of the presented study should be better worked out. 

Experimental section: 

-       Please improve the caption of Figure 1 and provide more details about the tensile testing. 

-       Please provide more information the preparation of the TEM specimens. 

Results and discussion: 

-       Please delete the following text “This section may be divided by subheadings. It should provide a concise and precise 97 description of the experimental results, their interpretation, as well as the experimental 98 conclusions that can be drawn.”

-       The description of the presented in Table 3 should be shortened and streamlined. The respective sections read rather confuse and tend to duplicate information. 

-       Figure 2, Figure 3 and Figure 4 should be combined and respective information should be presented in 1 figure only. 

-       The SEM micrographs presented in Figure 7 should be presented with scale bars only.

-       Figure 8: The right abbreviation for arbitrary units is arb. u. and not a. u. 

-       Please improve the quality of the EDX images. 

-       EBSD results are appreciated but no experimental details have been given. 

-       The part results and discussion is rather descriptive. There is no real discussion going on. This must be greatly improved to make the article suitable for publication.

Author Response

(The authors gave the same response as above.)

Reviewer 3 Report

Manuscript numbered “coatings-2083434” has been reviewed:

The introduction needs some improvements.

It is suggested to validate the DOE method before statistical analysis. For this purpose, predict the values of output with the Taguchi method and then compare it with the experiment.

Please add the fabrication process parameters in a table.

Why the error column in table 2 is empty?

Please present tensile diagrams or mathematical power-low models for them.

The following papers are suggested for the introduction section:

Lightweight design of an AlSi10Mg aviation control stick additively manufactured by laser powder bed fusion

High-cycle fatigue properties of curved-surface AlSi10Mg parts fabricated by powder bed fusion additive manufacturing

Effect of direct aging and annealing on the microstructure and mechanical properties of AlSi10Mg fabricated by selective laser melting

A review of Industry 4.0 and additive manufacturing synergy

Author Response

(The authors gave the same response as above.)

Round 2

Reviewer 2 Report

Thank you very much for revising the manuscript.

The authors have adequately addressed all raised points. 

Therefore, the article is recommended for publication in its current version. 

Reviewer 3 Report

The paper is ready to publish.